# Comparative Analysis of Blood Clot, Plasma Rich in Growth Factors and Platelet-Rich Fibrin Resistance to Bacteria-Induced Fibrinolysis

**DOI:** 10.3390/microorganisms7090328

**Published:** 2019-09-07

**Authors:** Tomas Puidokas, Mantas Kubilius, Donatas Nomeika, Gintaras Januzis, Erika Skrodeniene

**Affiliations:** 1Department of Oral and Maxillofacial surgery, Lithuanian University of Health Sciences, LT-46383 Kaunas, Lithuania; 2Department of Laboratory Medicine, Lithuanian University of Health Sciences, LT-46383 Kaunas, Lithuania

**Keywords:** platelet concentrate, fibrinolysis, dry-socket, d-dimer, bacteria

## Abstract

Alveolar osteitis (AO) is a common, painful postoperative complication after tooth extraction. Fibrinolytic activity in the extraction socket is one etiological factor. Platelet concentrates are used to prevent and treat AO. The aim of this study was to find out whether the positive effect of platelet concentrates can be related to resistance to bacteria-induced fibrinolysis. Blood from 45 human volunteers was used to prepare four media: blood clot medium as control group; PRF and PRGF first fraction (PRGF I) and PRGF second fraction (PRGF II) as study groups. Additionally, collected blood was used for blood plasma preparation on which evaluation of initial value of d-dimer concentration was performed. A solution of five different microbes (*Staphylococcus aureus*, *Streptococcus pyogenes*, *Streptococcus pneumonia*, *Bacillus cereus*, and *Candida albicans*) was adjusted to 0.5 McFarland (1 × 10^8^ CFU/mL) and then diluted to 0.25 McFarland (0.5 × 10^8^ CFU/mL). The d-dimer concentration was evaluated after one and three hours of bacteria exposure. The resistance to fibrinolysis was not statistically distinguished among any media groups at any time. *S. pneumoniae* was statistically active in PRF after three hours. *C. albicans* was statistically active in PRGF II after one hour and in PRF between the first and third hour and after three hours. *S. aureus* and *B. cereus* were statistically active in PRGF II after three hours. *S. pyogenes* was statistically active after one hour, between the first and third hour, and after the third hour in all groups. *S. pyogenes* was the most active bacterium. Different blood formulations were not distinguishable based on resistance to bacteria-induced fibrinolysis. Low fibrinolytic properties of the found major microbes suggests that bacteria-induced fibrinolysis is one of the leading causes of absence of a clot in a post-extraction socket to be clinically insignificant. The initial absence of a clot or its mechanical elimination during formation or the healing period are major causes of dry socket.

## 1. Introduction

Alveolar osteitis is a common, painful postoperative complication that occurs one to three days after tooth extraction [1]. Symptoms include severe pain in the extraction site, moderate to severe dull headache with pain occasionally radiating to the ears, halitosis, and dysgeusia or altered taste; symptoms can last up to 28 days [2]. The prevalence varies approximately 1–5% for all dental extractions and up to 30% for third molar extractions, depending on the degree of tissue trauma caused by tooth extraction [3] as well as predisposed risk factors such as smoking and poor oral hygiene [4,5]. The etiology of alveolitis is associated with partial or total loss of the blood clot [1]. Fibrinolytic activity in the extraction socket is one of the etiology factors associated with the clot loss or lysis and exposure of the bone to the oral cavity [1]. The increased fibrinolysis is thought to be attributed to surgery trauma, pathogenic fibrinolytics, and enzymatic changes due to inflammation and the presence of high amounts of bacteria before and after surgery [4,5].

The oral cavity is colonized by numerous bacteria including *Staphylococcus aureus*, *Streptococcus pyogenes*, *Streptococcus pneumoniae*, and *Candida albicans* [6,7,8]. The balance of this microflora contributes to active fibrinolysis which may induce blood clot lysis [9,10,11,12,13]. One fibrinolytic bacterium, *B. cereus*, is abnormal to the oral cavity and may originate from *B. cereus*-colonized food intake [13,14].

Numerous methods have been reported for prevention and management of AO [5]. One of the prevention approaches is the application of platelet concentrates such as plasma rich in growth factors (PRGF; BTI Biotechnology Institute, San Antonio, Spain) or platelet-rich fibrin (PRF). Both concentrates consist of various growth factors to boost the healing process [15,16]. The difference between PRGF and PRF is the white blood cell count: PRGF contains none [17]. Preparation of PRGF involves separation of the different fractions: the first (PRGF I) contains a platelet count similar to that of peripheral blood, the second (PRGF II) contains a higher quantity of platelets and growth factors [18]. PRGF II clinically is used as a clot to fill the extraction socket and PRGF I as a membrane for socket coverage [5,19]. Platelet concentrates are effective in reducing the incidence of AO [5,20,21]. Both the abovementioned platelet concentrates are effective for management of already present AO [17,22].

Platelet concentrates have an antibacterial effect. Positive activity of leukocyte- and platelet-rich plasma against *S. aureus*, *Enterococcus faecalis*, and *Pseudomonas aeruginosa* has been reported [23]. A 2016 review offered an overview of the antimicrobial effect of platelet concentrates on various microbes. PRF products exhibit an antibacterial effect on *Porphyromonas gingivalis* and *Aggregatibacter actinomycetemcomitans* [24]. Thus, PRF may have an antifibrinolytic effect due to leukocyte presence.

d-dimer is a specific degradation product of fibrin; its testing is used for the diagnosis of a variety of thrombosis-related conditions. There are three techniques for d-dimer evaluation: enzyme-linked immunosorbent assays (ELISAs), a whole-blood agglutination assay, and latex agglutination assays [25]. d-dimer value elevation indicates active fibrinolysis [26]. Certain factors influence d-dimer test values such as the extent of thrombosis, fibrinolytic activity, or the presence of active infection [27,28].

Although the leading factor of AO is the lysis of blood clot, no present studies on platelet concentrates were found considering the resistance to microbe-induced fibrinolysis. The resistance to fibrinolysis characteristic can be defined as reduced degradation of fibrin in the presence of fibrinolytic microorganisms. In this study, we examined the resistance of both PRGF and PRF to fibrinolytic bacteria (*S. aureus*, *S. pyogenes*, *S. pneumoniae*, *B. cereus*, and *C. albicans*) and compared the results with the blood clot medium.

## 2. Materials and Methods

The study was performed following the principles of the Declaration of Helsinki, as revised in 2008 and after approval from Kaunas Regional Biomedical Research Ethics Committee, Lithuania. Approval number: BE-2-15.

### 2.1. Preparation of Blood Plasma

Additional blood samples were collected from each group volunteer (45 total) using 18G needles. For each individual, two (5 mL each) tubes (containing 0.2 mL of sodium citrate) of peripheral blood were collected and immediately placed in a centrifuge for nine minutes, 2500RCF (Hettich Rotina 35, Tuttlingen, Germany). After three hours of incubation at room temperature, concentration of d-dimers of blood plasma was measured (STAGO Diagnostics, STA Compact, Asnières sur Seine, France). The first blood plasma d-dimer count was used as the initial value.

### 2.2. Preparation of Blood Clot

Blood samples were collected from 15 healthy volunteers (20–24 years old). For each individual, five (5 mL each) tubes of peripheral blood were collected. The tubes were kept at room temperature for 10 minutes and were exposed to 37 °C for three hours to ensure clot retraction because non-retracted clots are more prone to lysis [29,30]. After the blood was fully coagulated, the blood serum was removed using a Pasteur pipette.

### 2.3. Preparation of Plasma Rich in Growth Factors (PRGF)

Preparation of plasma rich in growth factors was conducted according to the manufacturer’s instructions. Blood samples were collected from 15 healthy volunteers (20–24 years old) using PRGF-Endoret^®^ Tubes (BTI Biotechnology Institute, S.L., Miñano, Spain) containing 0.2 mL of sodium citrate. For each individual, two tubes of peripheral blood were collected and immediately centrifuged at 580 g for 8 min. Fraction 1 (above Fraction 2) and Fraction 2 (the fraction 2 mL above the buffy coat) were collected and subjected to two separate tubes. Fraction 1 and Fraction 2 fibrin scaffolds were prepared by activating PRGF fractions with calcium chloride (Endoret Dentistry, Vitoria, Spain) at 37 °C for 1 h (50 μL of calcium chloride for each mL of PRGF). Next, each PRGF Fraction was aliquoted into five pieces and then transported to five different tubes for each fraction without additional reagents.

### 2.4. Preparation of Platelet-Rich Fibrin (PRF)

Blood samples were collected from 15 healthy volunteers (20–24 years old). For each individual, one tube of peripheral blood was collected and immediately placed in a centrifuge (Process for PRF, Nice, France). Centrifugation was performed according to the following protocol: A-PRF+, sterile glass coated plastic tube (10 mL; 1300 rpm for eight minutes). The PRF was then separated from the rest of the blood clot, divided into five parts (for each bacterium), and transported to five different tubes without additional reagents.

### 2.5. Preparation of Bacteria and Fungus Suspension

Our study used *S. aureus* ATCC 29213 culture grown in tryptic soy agar medium [31]. *S. pyogenes* isolates were cultured from clinical material collected from patients of the Lithuanian University of Health Sciences Kaunas Clinics with a suspected infection. Each subject underwent swab collections of oral cavity involving tongue and buccal mucosa. Isolates were cultured on a 5% sheep blood agar and confirmed to be pathogens by the laboratory of Department of Laboratory Medicine of Lithuanian University of Health Sciences. *S. pyogenes* was identified by a polymerase chain reaction (PCR) method using the Illumigene system (Meridian Bioscience, Inc., 2015, Cincinnati, OH, USA). Streptokinase (SK) assay protocol was used as described by T. T. Huang et al. [32]. The bacteria were streptokinase-positive. *S. pneumoniae* ATCC 6301 and *B. cereus* ATCC 12826 strains were used. *S. pneumoniae* was cultured on tryptic soy agar with sheep blood medium and *B. cereus* on a nutrient agar plate. *C. albicans* strain Ca60 was used which was obtained by a swab technique of oral candidiasis lesions. The fungi were grown in chromogenic HiCrome *Candida* medium (HiMedia, Mumbai, India) and confirmed via PCR. The ATCC cultures were obtained from ATCC Global Bioresource Center (Manassas, VA, USA).

After 24–48 h of incubation on growth media, colonies were harvested and transferred into a tube of saline. Optical density was adjusted to 0.25 McFarland (0.5 × 10^8^ CFU/mL) using saline. The solution was poured into disposable tubes, 1 mL each.

### 2.6. PRF, PRGF, and Blood Clot Fibrinolysis Activity Assay

Blood plasma in a quantity of 400 µL was carried to every PRF, PRGF, and blood clot tube using a Finnpipette batcher. The d-dimer count of blood plasma had been already measured before the application to study media (initial value). Next, using the same batcher, 600 µL of prepared bacteria suspension was carried to the same tubes. The concentration of d-dimers in the blood plasma of different media tubes was evaluated after one and after three hours [29,33] (STAGO Diagnostics, STA Compact, Asnières sur Seine, France) and compared with initial measurement.

### 2.7. Statistical Analysis

Results are presented as the mean, as appropriate. Given the assumption that the outcome is approximately not normally distributed, d-dimer differences were analyzed with use of nonparametric tests. Differences among the formulations were evaluated with use of the one-way ANOVA (analysis of variance) test, the Wilcoxon matched-pairs test, and Kruskal–Wallis test for multiple comparisons. A *p* value of <0.05 was considered significant for these tests.

## 3. Results

The results were structured into two categories. The first category was evaluated based on the resistance of each medium (blood clot, PRF, PRGF I, PRGF II) to bacteria-induced fibrinolysis and what the d-dimer concentration was in each specific group during the entire timescale. In the second category, bacteria and their fibrinolytic activity were evaluated by highest d-dimer concentration or its change in the presence of a microbe at a specific time.

First, the resistance to fibrinolysis was not statistically distinguished among all media groups in the initial stage or after the first or third hour. d-dimer concentration was not statistically diverse (*p* > 0.05). However, mean d-dimer concentration of PRF group was 1.02 ± 0.21µg/mL and 0.66 ± 0.12 µg/mL in the blood clot group in the presence of *C. albicans*. The mentioned values were statistically diverse (*p* < 0.05). Thus, PRF showed elevated d-dimer count in comparison with the blood clot group during the entire timescale, as seen in Figure 1.

Different types of bacteria expressed diverse fibrinolytic activity based on the medium and time of exposure. *S. pneumoniae* did not show a statistically significant elevation of d-dimer count after one and three hours in any media groups except for PRF after three hours. Mean d-dimer concentration of 0.89 ± 0.21 µg/mL in the PRF group after three hours was statistically significant, as seen in Figure 2 (*p* > 0.05). *C. albicans* had raised d-dimer concentration in PRGF II group after one hour and mean d-dimer concentration was held at 0.7 ± 0.21 µg/mL (*p* < 0.05). In the PRF group, between the first and third hour, d-dimer concentration elevated by 0.17 ± 0.09 µg/mL (*p* < 0.05), as seen in Figure 2. After three hours, d-dimer value of 0.96 ± 0.34 µg/mL was statistically significant (*p* < 0.05) only in the PRF, as seen in Figure 2. A comparison of the PRGF II and PRF groups in the presence of *C. albicans* identified that *Candida* bacterium is more active in the PRF medium (*p* < 0.05). *S. aureus* and *B. cereus* increased d-dimer concentration after three hours in PRGF II medium with values of 0.69 ± 0.28 µg/mL and 0.7 ± 0.32 µg/mL respectively (*p* < 0.05). *S. pyogenes* exhibited d-dimer raise in all groups after one hour. Mean concentration rise compared with initial value was 0.85 ± 0.83 µg/mL. Between the first and third hours, the mean d-dimer change in concentration was 0.83 ± 0.88 µg/mL. This change was statistically significant (*p* < 0.05). After three hours of exposure, d-dimer values were statistically higher in comparison with other bacteria. Blood clot d-dimer concentration was 1.6 ± 1.08, PRF was 1.85 ± 1.09, PRGF I was 1.49 ± 1.62, and PRGF II was 1.19 ± 0.3 µg/mL (*p* < 0.05), as seen in Figure 3. Thus, the observed results regarding *S. pyogenes*’ fibrinolytic activity indicates that this kind of bacterium is the most active in blood clot and platelet concentrates.

## 4. Discussion

Platelet concentrates have become a major factor in everyday oral surgery practice; accordingly, it is essential to evaluate each product as an anti-bacterial approach. Blood clot lysis is mostly dependent on microorganism fibrinolytic activity. Platelet concentrates undoubtedly provide an acceleration of tissue regeneration. Hence, it is important to assess fibrinolytic resistance of platelet products prepared by different protocols. We created and adapted a unique fibrinolysis measurement and evaluation protocol, including five fibrinolytic active pathogens in direct contact with the concentrates and blood clot as no comparative characteristics in vitro have been described previously. Clinical studies report that both PRF and PRGF are effective [5,17]. The question remains as to how the structure and composition of these concentrates may influence fibrinolytic resistance. Therefore, our study was performed via in vitro circumstances. Various studies have measured the level of fibrinolysis using spectrophotometry [34,35]; however, this assay requires the elimination of living circulating cells. Our objective was to create an actual in vitro model of blood clot and platelet concentrate interaction with pathogenic microbes. The d-dimer assay was chosen for the measurement of fibrinolysis level as it is sensitive and offers predictive value describing clot dissociation [36].

PRF and PRGF are two leading products which differ in both preparation protocol and biological composition. L-PRF includes leukocytes, fibrin mesh and platelets while PRGF consists mostly of platelets and fibrin [37,38]. Studies have describe the presence of leukocytes unevenly, with both pro-inflammatory or anti-inflammatory properties observed. Both in vitro studies and literature review studies support these discussions [39,40,41]. During our research, no significant difference in the rate of fibrinolysis was observed in all groups, which may indicate that the presence of leukocytes does not have any significant role in a product’s anti-fibrinolytic properties.

Furthermore, different bacteria expressed different fibrinolytic activity. As described in 2014 by [10], *S. pyogenes* presents a thorough fibrinolytic potential in blood clot alone and its streptokinase is highly specific for human plasminogen. Our results determined the same properties of *S. pyogenes* in both platelet concentrates, as the species stands out as the most fibrinolytic in all groups. These data may reveal that platelet concentrates are unable to prevent *S. pyogenes*’ secreted streptokinase-induced fibrinolysis and present the same anti-fibrinolytic properties as the blood clot alone. Nevertheless, McArthus et al. [42] stated that different allelic variations of *S. pyogenes* streptokinase express functional differences in plasminogen activation. This finding indicates that a variant of *S. pyogenes* strains may exhibit a lower or even higher fibrinolytic activity in comparison with our tested *S. pyogenes* strain. Streptokinase gene sequences (*ska*) are divided in to two groups: cluster type-1 and cluster type-2. The latter is further divided into type-2a and type-2b. Type-2b has shown reduced affinity to Glu-plasminogen. All SK variants activate plasminogen when an activation complex is formed together with plasmin. Additionally, type-2b and type-1 clusters are inhibited by α_2_-antiplasmin [43]. High fibrinolysis of *S. pyogenes* in our study can be linked to SK type-1 or type-2a cluster that is a cause of increased bacterium virulence [43]. Tewodros et al. [44] in their animal study identified additional *ska*3, *ska*4, *ska*7, and *ska*8 genotypes that exhibited none or very low plasminogen activation. *Ska*5, *ska*6, and *ska*9 were identified by Cleary and Cheng [45]. Authors found that the *ska*5 strain was not responsible for nephritis symptoms while *ska*6 and *ska*9 clusters were linked to acute post-streptoccocal glomerulonephritis. Nevertheless, these findings cannot be linked to our study as to which species of oral microflora was collected. Slow fibrin clot lysis by *S. pneumoniae* was identified by one study. The results were linked to possibly low plasminogen concentration [46]. Thus, our results can be explained in the same manner and fibrinolysis of *S. pneumoniae* may take longer than three hours to occur. Staphylokinase cause fibrinolysis only when it is bound to plasmin unlike streptokinase which forms an active complex with plasminogen. Additionally, staphylokinase is not considered a staphylococcal virulence factor of *S. aureus* [47]. *Candida* spp. are capable of adhering to platelets and fibrin [48]. Higher *C. albicans* activity in PRF and PRGF II medium can be linked to this precise characteristic. With regard to the PRGF I fraction, no active fibrinolysis can be observed due to a lower platelet count because PRGF I fraction has a similar platelet count as peripheral blood. Additionally, possible sodium citrate, which is used as an anticoagulant in PRGF preparation, is known to inhibit *C. albicans* adhesion to platelets [18,48]. A study by Ponnuswamy et al. [12] demonstrated how a fibrin net was digested in 60 minutes at room temperature in the presence of *B. cereus*. The effect was dependent on the enzyme concentration. In our study, *B. cereus* was statistically active only in the PRGF II group. Such a result may be explained by possible insufficient fibrinolytic enzyme presence. Nevertheless, bacteria induce secretion of cytokines from host cells which may act in concert with fibrinolytic systems [47]. Our study model does not contain a host environment and does not represent the extent of a clot fibrinolysis.

In the oral surgery field, platelet concentrates are routinely used for socket preservation and as an alveolitis prevention measure [49]. Numerous studies display significant results with regard to new bone formation, its quality and quantity, as well as reduced pain and healing time [50,51,52]. In consideration of the results obtained from our research, PRF, PRGF, and blood clot present no significant difference in anti-fibrinolytic characteristics, implying that the physical properties of fibrin mesh of different blood preparations were not distinct [53].

Bacteria- or fungi-induced fibrinolysis is described as one of the leading factors contributing to the absence of a clot in a post-extraction socket. In our study, *S. pyogenes* showed significant fibrinolytic activity and may be considered as a pathogen in AO development. Nevertheless, *S. pyogenes* exhibits different SK gene sequences that may determine decreased or even absent fibrinolytic capabilities. The low fibrinolytic properties of four major microbes found in the oral cavity disproves the statement that microorganism-induced fibrinolysis is a leading factor of clot absence in AO; this suggests that the initial absence of a clot or its mechanical elimination during formation or healing period are major causes of dry socket in practice. In clinical practice, it is essential to assure adequate post-extractional bleeding by using anesthetics without vasoconstrictors and mechanical protection of the wound. Nevertheless, fibrinolysis is dependent on bacterial activity and should not be in a state of neglect. Thus pre- and postoperative oral cavity treatment with antibacterial agents such as chlorhexidine is suggested to fully minimize the incidence of AO [54].

## Figures and Tables

**Figure 1 microorganisms-07-00328-f001:**
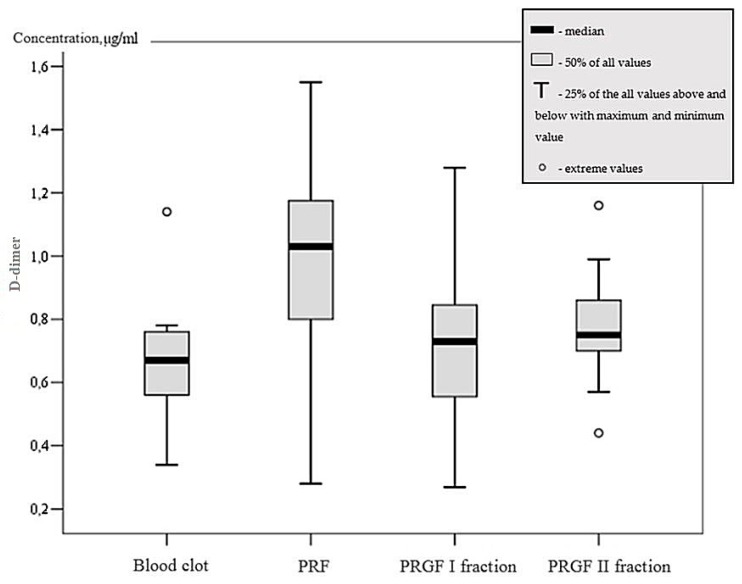
d-dimer concentration based on medium in *C. albicans* group.

**Figure 2 microorganisms-07-00328-f002:**
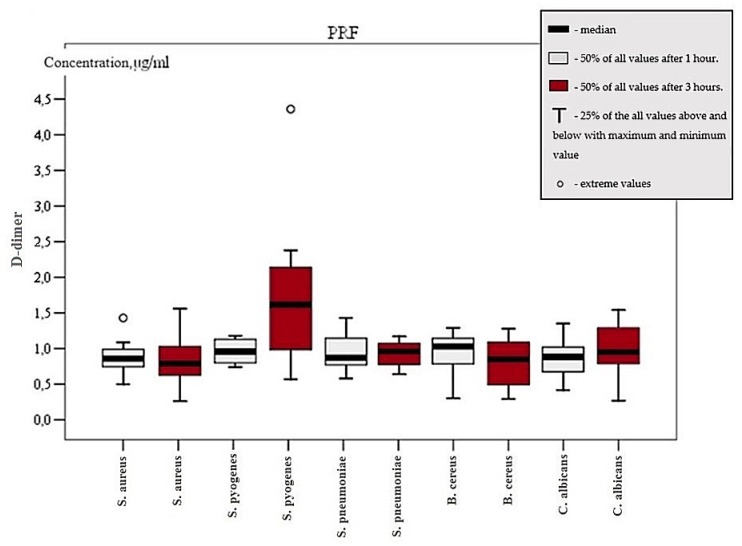
d-dimer concentration in PRF group between first and third hour.

**Figure 3 microorganisms-07-00328-f003:**
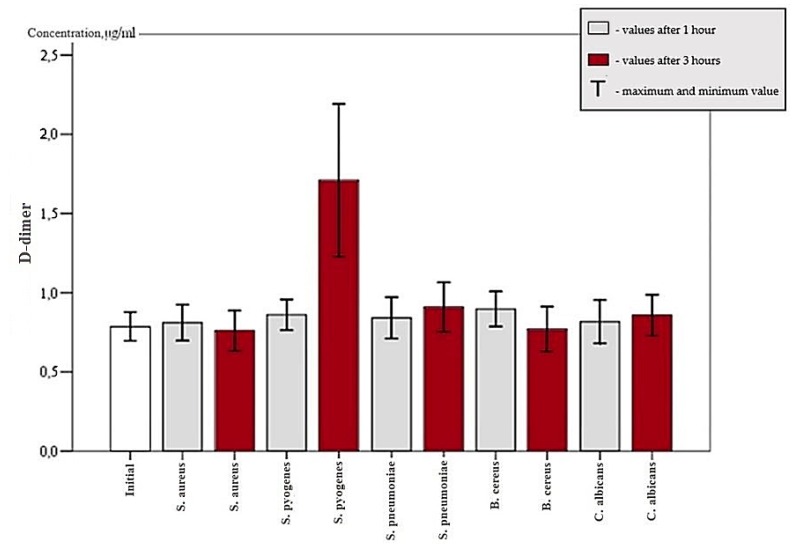
d-dimer concentration in all groups after one hour and after three hours.

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
