# Peer review of "Comparative Analysis of Blood Clot, Plasma Rich in Growth Factors and Platelet-Rich Fibrin Resistance to Bacteria-Induced Fibrinolysis"

_microorganisms, 2019, doi:10.3390/microorganisms7090328_

Round 1

Reviewer 1 Report

The current manuscript is improved.

Figure 2,3 and 4. The x-axis legends must be replaced with the proper scientific name of microorganisms. for example, Albicans can be replaced with C. albicans.

Author Response

Reviewer 1

Figure 2,3 and 4. The x-axis legends must be replaced with the proper scientific name of microorganisms. for example, Albicans can be replaced with C. albicans

Response: Figure 2 has been removed considering remarks of the second reviewer. Other figures have been adjusted regarding your statement.

Reviewer 2 Report

Puidokas et. al. should read the following suggesntions and review the current manuscript. It requires substantial work prior to publication:

Introduction

line 43 should read "The prevalence varies from 1-5% for all dental extractions up to 30% for third molar...."

In lines 45 to 47 use etiological or etiology once, otherwise it sounds too redundant.

In lines 51 - 54, even if mentioned and completely spelled out in the abstract, all genus and species names should be fully spelled out first time mentioned in the manuscript.

Line 59, "PRGF contains none [17]."  Please explain with more clarity, why is it important to test PRGF and PRF and PRGF II?

Sometime you refer to the disease AO and sometimes, even after defining AO, you refer to it as alveolar osteitis.  This can be left as is, but if I suggest you stick to the AO format so that it's easier and more consistent to follow.

I strongly suggest you fully explain D-dimer in the introduction. This is not a very common unit of measure, but very specific to blood clotting. The entire paper and evidence that tests the main hypothesis revolves around D-dimer units. To better evaluate the significance of the data, please take your time and dedicate a paragraph to this concept and what it means (high values vs low values).

Materials and Methods

Line 97: add a molar concentration for calcium chloride

Line 99: instead of saying "divided in five pieces" say "aliquoted" (assuming the five pieces are equal volumes).

Lines 77, 92, 102 you keep repeating 18G needles.  I think the first time was enough. Just mention that blood drawing what always done with this size needle.

Lines 114 tp 120. For every microbe that was purchased directly from ATCC, please say so. If the microbe was obtained from a different lab, mention the PI of that lab. If the authors collected the strains themselves, there must be a methodology to evaluate that this is (or these are) the correct strain(s) and not some other microbe.  There are several ways to do this but I think the most acceptable is PCR for either 16S or 18S followed by sequencing and BLAST.  If this was not done, I strongly suggest you do this to be sure what strains are being used in this study.

Line 117: authors state that colonies were touched with a sterile tag and transferred to saline.  This would not result in growth and the CFU concentration in the saline would be very low and not cloudy at all. A culture of 10^8 CFU/ml is fairly cloudy.  I assume authors first grew the microbes in some media and then adjusted the OD to yield 10^8 CFU/ml. Please describe and clarify.

Lines 124 - 125:  Because I am still ignorant about D-dimers, I am not sure why the authors chose 1 and 3 hours to measure their results.  Going along with this, the experimental design should include a negative control (blood clotting on its own) and a positive control (something known to be fibrinolytic).  All assays with microbes should be directly compared to these two controls. Please add these controls.

Results

Figure 1 should be the blood clot, the PRF and the PRGF fractions exposed to nothing (negative control) and to a known fibrinolytic agent, compared side by side.  On the Y axis, what is being measured? D-dimers? are there units for this? if so put that on Y axis labels. The legend for figure 1 and all other figures should tell the reader what this table is all about without having to read the text.

Medium is singular.  Media is plural.  Please change "mediums" to "media" and "medium groups" to "media groups".

Both Figure 2 and Figure 3 are testing the microbes on PRF. Figure 2 is only the 3 hour values but Figure 3 shows 1 hour and 3 hours. This means that the data Figure 2 are already included in Figure 3.  Figure 2 is not necessary.

What is Figure 4 showing? PRGF? or PRGF II?  What is "initial" in that figure? is that the blood clot? or the sample without any microbes?

From lines 146 to 164, please read this slowly and look at the data. I disagree with most of this. Either it is incorrect or I can't follow what you mean to say because it is unclear. From what I see, only S. pyogenes is yielding a significantly different result compared to all others in everything you have tested.

Discussion

Line 197: bacteria is plural, bacterium is singluar.  Rephrase to "...the species stands out..." or "...the bacterium stands out..."

Line 198: Data is plural.  "These data...."

Line 206: "...only when it is bound to..."

Line 208: "...not considered as a staphylococcal..."

Line 215 - 216: Not sure what you mean with the statement that is underlined. Why this is underlined?

Line 218 - 219: what should be sufficient time for clot lysis to occur? Design an experiment with sufficient time for clot lysis to occur and include it in this study.

Discuss further the different strains S. pyogenes and their streptokinases. This is the clearest future direction and deserves more attention.

Provide a final conclusion statement for your study. 

Author Response

Reviewer 2

All the mentioned remarks towards grammar and vocabulary were corrected (visible as “track changes” in the manuscript).

Introduction

I strongly suggest you fully explain D-dimer in the introduction. This is not a very common unit of measure, but very specific to blood clotting. The entire paper and evidence that tests the main hypothesis revolves around D-dimer units. To better evaluate the significance of the data, please take your time and dedicate a paragraph to this concept and what it means (high values vs low values).

Response: Information about D-dimer and its‘ significance has been added as a separate paragraph in the introduction section (Lines 73-78).

Materials and Methods

Line 97: add a molar concentration for calcium chloride

Response: Molar concentration of CaCl2 has not been added because the manufacturer does not provide such information. The reagent is a part of the PRGF kit.

Lines 114 tp 120. For every microbe that was purchased directly from ATCC, please say so. If the microbe was obtained from a different lab, mention the PI of that lab. If the authors collected the strains themselves, there must be a methodology to evaluate that this is (or these are) the correct strain(s) and not some other microbe.  There are several ways to do this but I think the most acceptable is PCR for either 16S or 18S followed by sequencing and BLAST.  If this was not done, I strongly suggest you do this to be sure what strains are being used in this study.

Response: Source of ATCC microbes has been added. The method for microbe cultivation and identification has been added (Lines 135-150).

Line 117: authors state that colonies were touched with a sterile tag and transferred to saline.  This would not result in growth and the CFU concentration in the saline would be very low and not cloudy at all. A culture of 10^8 CFU/ml is fairly cloudy.  I assume authors first grew the microbes in some media and then adjusted the OD to yield 10^8 CFU/ml. Please describe and clarify.

Response: The information regarding bacteria suspension preparation has been clarified in the text (Line 149).

Lines 124 - 125:  Because I am still ignorant about D-dimers, I am not sure why the authors chose 1 and 3 hours to measure their results.  Going along with this, the experimental design should include a negative control (blood clotting on its own) and a positive control (something known to be fibrinolytic).  All assays with microbes should be directly compared to these two controls. Please add these controls.

Response: The chosen time of measurement was based on previous in vitro whole blood clot lysis studies (Line 156). Blood clot group itself is a control group in our study. In daily practice of oral surgery after tooth extractions blood clot serves as a biological barrier and a framework of potential bone recovery. Platelet concentrates are used as an autologous material with advanced features of a blood clot. We measured the D-dimer concentration change of the blood plasma (Lines 153-154). The primarily obtained blood plasma was poured in the tubes with blood clot, PRF and PRGF fractions and after incubation was extracted and measured again for the evaluation of the increased D-dimer. Additional information added in Lines 152-157 regarding blood plasma use in our study.

Results

Figure 1 should be the blood clot, the PRF and the PRGF fractions exposed to nothing (negative control) and to a known fibrinolytic agent, compared side by side.  On the Y axis, what is being measured? D-dimers? are there units for this? if so put that on Y axis labels. The legend for figure 1 and all other figures should tell the reader what this table is all about without having to read the text.

Response: The presentation of figures has been improved based on both reviewer remarks. Blood clot in our study serves as a control group. A thorough explanation was presented above.

Both Figure 2 and Figure 3 are testing the microbes on PRF. Figure 2 is only the 3 hour values but Figure 3 shows 1 hour and 3 hours. This means that the data Figure 2 are already included in Figure 3.  Figure 2 is not necessary.

Response: Figure 2 has been removed.

What is Figure 4 showing? PRGF? or PRGF II?  What is "initial" in that figure? is that the blood clot? or the sample without any microbes?

Response: Figure 4 is showing all groups (Blood clot, PRF, PRGF I/II) together as mean values. The initial value is of the blood plasma which is measured prior to exposing study groups to bacteria. The alongside presented values are values of blood plasma which is removed from tubes of blood clot, PRF and PRGF I/II after 1 and 3 hours of incubation and measured for D-dimer concentration. The clots of Blood, PRF and PRGF I/II were left as it is. The blood plasma was our aim of D-dimer measurement.

From lines 146 to 164, please read this slowly and look at the data. I disagree with most of this. Either it is incorrect or I can't follow what you mean to say because it is unclear. From what I see, only S. pyogenes is yielding a significantly different result compared to all others in everything you have tested.

Response: Lines 188-212 were edited.

Discussion

Line 218 - 219: what should be sufficient time for clot lysis to occur? Design an experiment with sufficient time for clot lysis to occur and include it in this study.

Response: The sufficient time of 1 and 3 hours was based on two studies. References presented in Line 154. Lines 282-283 were edited.

Discuss further the different strains S. pyogenes and their streptokinases. This is the clearest future direction and deserves more attention.

Response: Different cluster variantions of streptokinase were discussed (Lines 257-267).

Provide a final conclusion statement for your study.

Response: Conclusion has been improved.

Round 2

Reviewer 2 Report

Line 143. It says S. Pneumoniae.  Should be S. pneumoniae.

Lines 148 - 150.  I still think that there was a growth period in broth before the saline steps.  CFU of 10^8/ml is too high to achieve from just resuspending colonies in saline.   

Erase the "C. albicans" label from Figure 1 on Y axis.  Instead of that, put the ug/ml of the D-dimer value. When you say "concentration ug/ml" in figure 1 and 2, what is the concentration of? put that in the Y axis.

Author Response

Line 143. It says S. Pneumoniae.  Should be S. pneumoniae.

Line 143. Corrected.

Lines 148 - 150.  I still think that there was a growth period in broth before the saline steps.  CFU of 10^8/ml is too high to achieve from just resuspending colonies in saline. 

Lines 148-150. Growth period duration on the media has been added, statement edited.

Erase the "C. albicans" label from Figure 1 on Y axis.  Instead of that, put the ug/ml of the D-dimer value. When you say "concentration ug/ml" in figure 1 and 2, what is the concentration of? put that in the Y axis.

Y axis title of Figure 1 and 2 has been corrected.

This manuscript is a resubmission of an earlier submission. The following is a list of the peer review reports and author responses from that submission.

Round 1

Reviewer 1 Report

The rationale for this work is to compare several bacteria that are present in the oral cavity  and which have fibrinolytic activity against various different plasma preparations for “fibrin resistance” and bacteria-induced fibrinolysis. Fibrin D-dimer levels were used as the output. The irationale  of this work is to investigate the cause of dry socket following tooth extraction and whether different plasma preparations offer treatment benefits

The paper needs further elaboration.

Major point

1.       Do the Staph .aureus and Strep. pyogenes strains tested actually express streptokinase and staphylokinase, respectively?. There is no information about the provenance of the bacteria tested.  There is variability as to whether strains of these two species express these zymogen activators -  the genes concerned are carried on mobile genetic elements. The authors should compare several Sak+/- strains

2.       Presentation of data in Figures - the legends are incomplete. An explanation for the boxes, whiskers and circles is required.

Minor points

3.       What is the rational for measuring D-dimers?. This is not apparent to the uninitiated

4.       It is not clear what is meant by fibrin resistance?

5.       The abstract should express a clear rationale for the study at its beginning.

6.       The English usage in the abstract  needs careful attention. Adhere to conventions regarding bacterial names - not  S. Aureus  (line 49)

Reviewer 2 Report

In this manuscript "Comparative analysis of Blood clot, Plasma rich in growth factors and Platelet rich fibrin resistance to bacteria induced fibrinolysis” Tomas Puidokas et.al., explored the resistance of blood clot, plasma rich in growth factors and platelet-rich fibrin to fibrinolysis induced by different pathogens, and assessed the activity levels of these microorganisms

The comments and suggestions for this manuscript are as follows-

1.      The author must include the figure legends (Figure 1,2,3 and 4).

2.      The author should be careful in writing the name of the bacterial strain, they must maintain the scientific nomenclature uniformly throughout the manuscript.

3.      In figure 1,2 and 3, the author should explain what these small circles “o” denote?

4.      Figures 3 and 4 presentations are not standard. The author should present in the group (1hours and 3hours) with proper statistics.

5.      Page 1 line 23 and page 3 line 105-106. The author should explain how he calculated 0.25 McFarland is 1x104 CFU/ml while stating 0.5 McFarland is 1x108 CFU/ml ? is it simple dilution? Calculation showing the 4log10 difference.

6.      The authors should provide a more comprehensive discussion and conclusion for the study.